# A Dual-Path Cross-Modal Network for Video-Music Retrieval

**DOI:** 10.3390/s23020805

**Published:** 2023-01-10

**Authors:** Xin Gu, Yinghua Shen, Chaohui Lv

**Affiliations:** School of Information and Communications Engineering, Communication University of China, Beijing 100024, China

**Keywords:** retrieval, video music retrieval, common representation spaces, emotion

## Abstract

In recent years, with the development of the internet, video has become more and more widely used in life. Adding harmonious music to a video is gradually becoming an artistic task. However, artificially adding music takes a lot of time and effort, so we propose a method to recommend background music for videos. The emotional message of music is rarely taken into account in current work, but it is crucial for video music retrieval. To achieve this, we design two paths to process content information and emotional information between modals. Based on the characteristics of video and music, we design various feature extraction schemes and common representation spaces. In the content path, the pre-trained network is used as the feature extraction network. As these features contain some redundant information, we use an encoder–decoder structure for dimensionality reduction. Where encoder weights are shared to obtain content sharing features for video and music. In the emotion path, an emotion key frames scheme was used for video and a channel attention mechanism was used for music in order to obtain the emotion information effectively. We also added emotion distinguish loss to guarantee that the network acquires the emotion information effectively. More importantly, we propose a way to combine content information with emotional information. That is, content features are first stitched together with sentiment features and then passed through a fused shared space structured as an MLP to obtain more effective fused shared features. In addition, a polarity penalty factor has been added to the classical metric loss function to make it more suitable for this task. Experiments show that this dual path video music retrieval network can effectively merge information. Compared with existing methods, the retrieval task evaluation index increases Recall@1 by 3.94.

## 1. Introduction

Video, a form of data that records life and conveys opinions, is produced and uploaded to the Internet by many users. Video is a media message that contains visual and auditory information. Among them, visual information is mainly presented in the form of moving images, while auditory information includes voice, ambient sound, background music, etc. Music, as an art form to express emotions, is often used to enhance the artistic effect of the video, resonate with the user and bring a better experience to the user. However, manually selecting the right music for a video is an extremely lengthy and laborious task. Short video applications often recommend music based on its popularity, and only part of the music conforms to the video. Therefore, the automatic retrieval of suitable music clips based on video information is a problem worth investigating. The study of this problem is equally beneficial in helping us to understand how art communicates between different modalities. Matching music with a given video belongs to the retrieval task, that is, the corresponding music information is retrieved through the video information. Some studies [1,2,3] suggest that both modals have semantic information related to the content. These studies therefore tend to take a self-supervised approach, using matched video and music pairs with no additional tagging information. They extract features using a generic network pre-trained on large datasets, which results in features containing certain redundant information. Furthermore, due to the diversity of visual content and the richness of musical emotional information, there is some asymmetry between the two modal data. Some research [4,5,6] considers that the emotional information of music is relatively rich and uses the emotional label to restrict the common space. However, they only use general networks to pre-train on emotional datasets, so that the network has the ability to learn emotional representations. This method has difficulty learning effective emotional information.

In this paper, we combine emotional information and content information to design a dual path cross-modal retrieval network. In the content path, a content common space with a fine network structure is designed and additional constraints are added to obtain the content sharing representation of video and music. In the emotional path, the emotional feature extraction scheme is designed to obtain emotional features. The representation of the emotional sharing of video and music is achieved by designing a network structure with simplified emotional common space. Combined with content and emotion, the common fusion space is designed, and the final representation features of video and music are achieved. By computing the similarity between features, the video music retrieval task is finished. Our main contributions are as follows:(1)A video-music retrieval dataset has been constructed and will be published on the website. The dataset is derived from films with a certain level of popularity, and each video–music pair includes an emotional descriptor and polarity labels.(2)A dual path video–music retrieval network combining content information and emotional information is designed. It can effectively learn various information and use this information to perform retrieval tasks.(3)A more task-consistent metric loss function was designed and used. By adding penalty factors to the data pairs, the metric loss function is optimized differently for different data pairs, achieving a dynamic optimization of the objective.

## 2. Related Work

In Section 2, we focus on research advances in video music retrieval and research work on emotion information in video and music. In particular, Section 2.1 details the current state of video music retrieval, based on two research ideas: content information and emotional information. Furthermore, we again emphasize that sentiment information can act as a bridge between the two modalities, video and music. Therefore, Section 2.2 focuses on the research on video emotion recognition and music emotion recognition, which helps to understand how sentiment information can be extracted from the different modalities.

### 2.1. Cross-Modal Video Music Retrieval

According to video retrieval, music belongs to the cross-modal retrieval technology, where cross-modal retrieval is able to retrieve information from one modality to the other modality associated with it. However, different modal data present heterogeneous low-level features, and only matching different modal data present semantically related high-level features. Faced with such a problem, cross-modal retrieval techniques often work by learning a common/shared representation space in which the semantic similarity of different modal data can be measured. In video music retrieval, the main current research approaches can be divided into content-based and emotion-based video-music retrieval.

Content-based video-music retrieval considers that both modalities have semantic content-related information, such as the consistency of the events of the video and the theme of the music, etc. Consequently, they tend to be self-supervised methods, using paired video and music pairs without additional label information. Hong S et al. [1] proposed the Cbvmr model, which uses traditional machine learning methods to extract the content features, and uses triple loss to limit the formation of the common representation space. Based on the Cbvmr model, Pretel L et al. [2] used a deep learning network model to replace the method of manually extracting features. Yi J et al. [7] propose a CMVAE model using textual information. Pretet L et al. [3] proposed and verified the effectiveness of movement information. The model uses Variational AutoEncoders (VAE) [8] as a common representation space and is optimized by generative loss functions. Zhang et al. [9] designed a VAE-CCA network combining the encoder–decoder structure. The network utilizes the decoder to reconstruct the features from the encoder, ensuring that the features reflect the structure of the original data, thus improving retrieval performance. Surís et al. [10] gave more attention to contextual long-term information. They used CLIP [11] to extract visual information from video frames, used Inception [12] to extract musical information from music segments. Lastly, the sequence information is entered into the common representation space structured by Transformer [13] to find the common features.

Emotion-based video–music retrieval considers that music is often used as media information to express emotions, so the emotional information of music is relatively rich. Aiming for that characteristic of music, research often uses emotional information to restrict the common representation space. Shin K et al. [4] used linear regression to predict the emotion of video and music, and compute the Euclidean distance between video and music data. Using emotional labels, Zeng D et al. [5] proposed a supervised DCCA (S-DCCA) model as a common representation space. Li B et al. [6] trained video networks and music networks separately on emotion datasets, making those networks to learn emotional information. Shang L et al. [14] believed that the connotative association of images and music included lyrical emotions in addition to semantic concepts, so they developed a connotation-aware music retrieval framework (CaMR).

### 2.2. Video, Music and Emotion

As two different modalities, video and music have a large heterogeneity of data. Video often contains a mix of content information such as actions, objects and scenes. In contrast, music, as a kind of audio with lyrical characteristics, contains more emotional information. Based on this heterogeneity, it is effective to take emotional information into account in video–music retrieval. There is a broad base of research on emotion information for video and music, namely video emotion recognition and music emotion recognition. Video emotion recognition (VER), the main research method for understanding video content, is not limited to facial information in emotion recognition, but focuses more on the emotional information conveyed by the video [15]. Musical emotion recognition (MER) has been a key area of research. Due to the specific nature of music, people can use music to resonate with their emotions and relieve stress.

In video emotion recognition, the effective selection of emotion frames [16] (frames rich in emotion information) is often the most fundamental part due to the redundant information and sparse emotional content of video frames. Emotion frame selection is mainly divided into key frame-based and video frame weight-based approaches. Key frame-based methods often use uniform sampling or clustering to select some video frames according to certain rules, which is fast, but not detailed enough for the selection of emotion frames. Methods based on video frame weights obtain video frames by calculating the weight value of each video frame [17,18,19], and video frames with larger weights contribute more to video emotion. In video emotion recognition, researchers in recent years have used deep learning networks to extract video features, including: action, object, scene and other features. These deep features contain rich semantic information and contextual content, with strong representation capability, and can effectively represent video emotion. In music emotion recognition, timbre features are often used in current research, and timbre features include zero crossing rate ZCR [20], acoustic spectrograms, spectral fluxes, etc. Among them, the acoustic spectrogram includes the Merle cepstral coefficient MFCC [21,22], and the cochlea spectrogram [23,24]. MFCC has information in the time-frequency domain of music, is proposed based on human speech and hearing. It has the characteristics of strong recognition ability and noise immunity, and the cochlea spectrogram contains richer texture information [25].

There are two main types of deep learning network structures used to extract features and recognize emotions in VER, MER: single networks and combined networks. The single network mainly uses CNNs to complete video feature extraction and feature to emotion space mapping. For example, 3DResNet [26] for video emotion recognition, 2DResNet [27] and Vggish [28] for music emotion recognition. In addition, due to the excellent performance of the Transformer structure in natural language, some researchers have applied this structure to vision, such as ViViT [29] and Video Swin Transformer [30] for video recognition. Combinatorial networks are mainly CNN-RNN [17,31], and long and short term memory networks (LSTM) are dominant in RNN.

## 3. Methodology

In this section, based on theoretical foundations and the current state of research, we first describe our proposed dual-path video music retrieval network (DPVM). The overall structure of our proposed DPVM is shown in Figure 1. Next, the architectures of the content public network, the emotion public network and the fusion public network are presented, respectively. Finally, adaptive improvements made to existing metric loss are presented.

### 3.1. The Network Overview

Video music retrieval differs from other cross-modal retrieval techniques in that it does not use semantic content information as the only bridge to cross-modal retrieval. The reason for this significant difference is mainly due to the specificity of music, which tends to express lyrical emotions as its main content. This difference leads to a different technical route for video–music retrieval tasks than for other cross-modal retrieval tasks. In the image–text retrieval task [32], the key point is how to accurately obtain the subject content information and background information of the image; and how to accurately obtain word information such as nouns, verbs and adjectives of the text, and to correspond these two modalities of information. In the video–text retrieval task [33], as video is dynamic information consisting of multiple images, it includes, in addition to the above key points, the accurate acquisition of temporal content information of the video. For the audiovisual correspondence and audiovisual matching tasks [34], which place more emphasis on video–audio correspondence, similar to the video–text retrieval task, the key point is also the acquisition of audio content information (e.g., train whistle, water flow, bird song). As these modalities contain more specific information and clear correspondence between different modalities, cross-modal retrieval techniques are often based on the consistency or correspondence of semantic content information as a criterion. Music, as a media message that expresses emotion, does not have as its primary purpose the conveyance of some explicit message. A positive melody is often enough to inspire the listener, a sad track often enough to express condolences. Based on the characteristics of music, both emotional and semantic content information are crucial in the task of video–music retrieval.

In current content-based video–music retrieval tasks, generic networks pre-trained on large datasets are often used to extract video and music features, and thus the features contain a large amount of redundant information, which leads to a misalignment of video and music feature information. For example, video features often contain a mishmash of content information such as actions, objects and scenes. Music, on the other hand, often contains information that is not as clear as video, and as a kind of audio with lyrical features, in addition to more specific content information, music also contains richer emotional information. Current emotion-based video-music retrieval tasks often use generic networks pre-trained on emotion datasets as a way to make the network capable of learning emotion representations. This approach does not effectively learn emotion information and still contains a large amount of redundant information.

Based on the above theoretical foundation and the current state of research, we use a combination of content information and emotion information to solve the video–music retrieval task, and designs a dual-path cross-modal retrieval scheme. For the content information of video and music, content features are extracted using a generic network model pre-trained on a large-scale dataset. The content features are refined using an encoding–decoding structure because of their mixed information and redundancy. An emotion extraction network is designed to extract features for the emotion information of videos and music, where the emotional discriminatory loss is used to optimize the feature extraction network. Compared with the content features extracted by the generic network model, the emotion feature information is more specific and less redundant, so the emotion common space with fully connected layer structure is designed to eliminate the heterogeneity of the two modalities and obtain the emotion shared representation. Finally, combining content information and emotion information, the fusion shared space is designed to obtain the final representation features of video and music, and the video and music retrieval task is completed by calculating the similarity between the representation features.

### 3.2. The Content Common Network

As shown in Figure 2, for video, we use the pre-trained 3DResNet on the Kinetics-600 dataset [35] to extract its content features. For music, we use the pre-trained Vggish on the AudioSet to extract its content features. In the design of the content common space, an encoder–decoder structure is adopted. The encoder reduces the dimension, and unifies the dimension of video and music features while sharing weights. The decoder is used to reconstruct the dimension. At the same time, it ensures that important information about dimensionality reduction features is not lost.

There are two parts which compose the optimization loss for content common networks: reconstruction loss and metric loss. Reconstruction loss optimizes encoder dimensionality reduction processing for content features. Metric loss optimizes the consistency of common feature information for generated video and music content.

Given video feature $V_{i}$ and music feature $M_{i}$. The reconstruction loss:(1)$$L_{R}=∥V_{i}-D\left(E\left(V_{i}\right)\right)∥+∥M_{i}-D\left(E\left(M_{i}\right)\right)∥.\;$$

The metric loss:(2)$$L_{M}=\frac{1}{2}\left(1-Y\right)∥E\left(V_{i}\right)-E\left(M_{i}\right){∥}^{2}+\frac{1}{2}\left(Y\right){\left\{max\left(0,margin-∥E\left(V_{i}\right)-E\left(M_{i}\right)∥\right)\right\}}^{2}.$$

The content loss:(3)$$L_{Content}=\lambda_{1}L_{R}+\lambda_{2}L_{M}.\;$$
where ∥·∥ denotes the cosine distance, $E\left(\cdot \right)$ denotes the encoder, and $D\left(\cdot \right)$ denotes the decoder.Y is paired labels, and if $\left\{V_{i},\;M_{i}\right\}$ is matching data, then Y = 0, otherwise Y = 1. The margin represents the distance threshold, and $\lambda_{1}$ and $\lambda_{2}$ represent weight coefficients.

### 3.3. The Emotion Common Network

As shown in Figure 3, because the emotional content of the video is sparse, that is to say the information of the video frame is redundant, and only a few frames contain rich emotional information. To efficiently get the emotional information from the video, we first choose the emotional key frames [36] from the video. Second, in the feature extraction network, the ViViT visual transformer model is selected. This not only improves the running speed, but also obtains the temporal context information of the video. In order to effectively obtain emotional information from music, the ResNet101 is adopted and the channel attention mechanism is introduced. Design an emotional common space with MLP as the structure.

There are two parts that compose the optimization loss for emotion common networks: distinguish loss and inter-modal loss. The distinguish loss adopts the cross-entropy loss function.
(4)$$L_{D}=-\sum Y\left(V_{i}\right)\log\frac{\exp\left(F\left(V_{i}\right)\right)}{\sum_{i=1}^{n}\exp\left(F\left(V_{i}\right)\right)}-\sum Y\left(M_{i}\right)\log\frac{\exp\left(F\left(M_{i}\right)\right)}{\sum_{i=1}^{n}\exp\left(F\left(M_{i}\right)\right)}.$$
where *Y*(∙) is the real emotion label, and *F*(∙) denotes the emotion common representation space of the multilayer perceptron structure.

Inter-modal loss preserves the difference in emotion characteristics of various modal data. This benefits the emotional common space to form pure emotional distinguish information from video and music.
(5)$$L_{M}=-∥V_{i}-M_{i}∥.\;\;\;\;$$

The emotion loss:(6)$$L_{Emotion}=\mu_{1}L_{D}+\mu_{2}L_{M}.\;$$
where $\mu_{1}$ and $\mu_{2}$ represent the weight coefficients.

### 3.4. The Fusion Common Network

The splicing fusion features are obtained by splicing two different features of the same modal data. As the similarity between the data is calculated, the information in the corresponding dimensions is calculated and accumulated separately. The splicing and fusion strategy can therefore be seen as a simple superposition between the similarities of the two feature dimensions. Therefore, on the basis of the splicing fusion, we have designed the fusion common space, where the splicing features obtain interactive fusion features through the fusion public space. This common space not only ensures that the heterogeneity of content and emotional information is further eliminated, but also that content and emotional information interact with each other, hence the choice of the fully connected layer (FC) in the design of the network structure. In FC, each neuron is formed with the participation of all neural units in the previous layer. This allows for comprehensive information transfer between features, thus ensuring that content information and emotional information can fully interact with each other.

The loss function optimized for common fusion space is the loss of contrast between interactive fusion features:(7)$$L_{Fusion}=∥F\left(V\right)-F\left(M\right)∥.$$

Combining Equations (3), (6) and (7), we obtain the objective function of the proposed method DPVM as:(8)$$\mathrm{Loss}=k_{1}L_{Content}+k_{2}L_{Emotion}+k_{3}L_{Fusion}.$$

### 3.5. The Polarity Penalty Metric Loss

In metric loss, the model is optimized by increasing the similarity between related data pairs and reducing the similarity between unrelated data pairs [37]. To some extent, there are some unrelated constraints and low relevance. In a batch with N pairs of data, {$v_{k}$,$m_{k}$} represents paired video–music data, and *φ* ($v_{k}$,$m_{i}$) is the similarity between the data. We use the cosine similarity Equation (9) to calculate *φ* ($v_{k}$,$m_{i}$), where *x*,*y* represents the sample data and n represents the data dimension.
(9)$$\cos\left(x,y\right)=\frac{\sum_{i=1}^{n}\left(x_{i}\times y_{i}\right)}{\sqrt{\sum_{i=1}^{n}{\left(x_{i}\right)}^{2}}\times \sqrt{\sum_{i=1}^{n}{\left(y_{i}\right)}^{2}}}.\;$$

The metric loss:(10)$$L_{\varphi \left(v,m\right)}=\sum_{k}^{N}\sum_{i\neq k}^{N}\left(\varphi \left(v_{k},m_{i}\right)-\varphi \left(v_{k},m_{k}\right)\right).$$

If *φ* ($v_{k}$,$m_{i}$) < *φ* ($v_{k}$,$m_{k}$), then $v_{k}$ and $m_{i}$ are called low-similarity data pairs. Repeatedly processing these low-similarity data pairs will cause the optimization function to focus on irrelevant targets for many times, resulting in low optimization efficiency. Therefore, we propose a more efficient approach based on the existing loss function.

Firstly, with regard to the issue of redundancy of the metric weight loss coefficient, the metric loss coefficient $\rho_{ki}$ of the low similarity data pair is set to 0.
(11)$$\rho_{ki}=\left\{\begin{array}{c} 0,\varphi \left(v_{k},m_{i}\right)<\varphi \left(v_{k},m_{k}\right)\\ 1,\;\;\;\;\;\;\;\;\;\;\;\;\;\;\;\;\;\;\;\;\;\;\;\;\;\;otherwise \end{array}\right\}.\;$$

Given the low weight coefficient and task fitness, and the issue of label difference is not taken into account, a polarity penalty coefficient $P_{ki}$ is proposed.
(12)$$P_{ki}=\left|label_{k}-label_{i}\right|.$$

The polarity penalty metric loss as:(13)$$L_{\varphi \left(v,m\right)}=\sum_{k}^{N}\sum_{i\neq k}^{N}\left(P_{ki}\rho_{ki}\varphi \left(v_{k},m_{i}\right)-\varphi \left(v_{k},m_{k}\right)\right).$$

## 4. Experiments

In this section, we first describe the dataset and implementation details used in the experiments. We then compare our evaluation results with other methods to demonstrate the effectiveness of our proposed approach. Finally, we perform a number of ablation studies to validate the performance benefits that each module of our method brings.

### 4.1. Dataset and Evaluation Metrics

Music compositions are protected by copyright, making datasets difficult to publish. Hong S et al. [1] obtained the Hong-Im Music-Video 200K (HIMV-200K) benchmark dataset from the large-scale tagged video dataset YouTube-8M [38]. The videos in this dataset were sourced from the YouTube website, and each video contains audio and video modalities. The videos are divided into 24 themes based on visual information. The data related to “music videos” under the “Arts & Entertainment” theme was selected, with approximately 200K video–music pairs, including official music videos, parody music videos and user-generated videos with background music. The author of the dataset provided the URLs of the YouTube videos under the “Music Videos” category. However, the HIMV-200K dataset suffers from the following problems: variable video quality, predominantly dance-themed music videos (MVs), noisy background music containing ambient sounds, and some video links not working. Due to the video link failure issue, the data available is 50K [2] in 2021 and 20K [10] in 2022.

For this purpose, we select well-known cinematic works from the film platform and capture video clips that contain only background music. We modify these clips to ensure that the duration of the video is about 15 s, forming a MVED dataset of about 3k video-music pairs. Emotional descriptions like “sad, happy, scared, surprised” and other adjectives were also provided. Based on these emotional descriptors, the dataset provides three polarity labels: “positive, negative, neutral”. We combine emotion descriptors and emotion annotations to form a compound tag of adjective–emotion polarity, such as happy–positive, sad–negative, etc. For each video data, five matching music tracks were selected based on the composite tags. The matching principles were as follows: priority was given to background music from the same film or TV series, the composite tags were consistent, and the subjective feelings of the video data and music data were consistent. Finally, five music data were matched for each video, forming a 10k video–music pair, which was divided into a training set and test set in the ratio of 7:3. More importantly, we publish the original dataset in a text table format that can be easily used by other researchers. The published form of the MVED dataset is shown in Table 1.

Following the previous works, we use Recall@K as an evaluation metric, as it is widely used in retrieval tasks.
(14)$$\mathrm{Recall}@K=\frac{\mathrm{TP}@K}{\mathrm{TP}@K+\mathrm{FN}@K}.\;$$
where TP@K represents the number of positive samples predicted to be positive in the first K results returned. FN@K illustrates the number of positive samples predicted as negative samples in the first K results returned.

### 4.2. Implementation Details

Due to the difference between the content shared path network and the emotion shared network path, the data formats used as input to the different paths are also different. In the content sharing network, each video data is first divided into six segments of the same number of frames in frame order (time) using a fixed frame sampling scheme, and 16 consecutive frames are selected in each segment. The input video datum is $V_{C}\in R^{6\times 16\times 3\times 256\times 256}$. Each music datum is extracted with MFCC features through the Librosa library, where the feature vector dimension is set to 32, the sampling frequency to 14,400, the FFT window length to 2048 and the step size to 2048, i.e., no overlapping samples. To ensure consistent music input dimensions, the maximum length was set to 192 by copying the data overlay operation. The input music data is $M_{C}\in R^{1\times 32\times 192}$.

To further reduce the redundancy of the video data in the emotion-sharing network, frames that contribute more to the emotion are selected, so the emotion keyframe scheme is chosen. The emotion keyframes scheme first detects the shot boundaries of the video based on the color histogram differences between frames and divides the shots one at a time. For each shot boundary frame, an emotion score is obtained by using a picture emotion recognition network model, setting an emotion score threshold and filtering out key shots. In each key shot, the frame with the largest inter-frame difference is selected as the key frame, which makes the difference between frames greater and further reduces redundancy. The final keyframe was set to 16, and the input video datum is $V_{E}\in R^{16\times 3\times 224\times 224}$. Each music datum was extracted from the Librosa library with MFCC features, where the feature vector dimension was set to 32, the sampling frequency to 16,000, the FFT window length to 1024 and the step size to 512, i.e., the overlap sampling rate was 50%. To ensure consistent music input dimensionality and subsequent network processing, the maximum length is set to 4096 dimensions by copying the data overlay operation. In this paper, it is segmented and evenly divided into 16 segments. This means that the input music datum is $M_{C}\in R^{16\times 32\times 256}$.

We use Adam optimizer [39] with {β1 = 0.5, β2 = 0.999} and train the model for 100 epochs. The learning rate is set to 1 × 10^−4^, and the batch size is 64. Metric loss uses the calculation method of cosine distance. In (3), (6) and (8), $\lambda_{1}$ = 0.8, $\lambda_{2\;}$= 1.0; $\mu_{1}\;$= 0.8, $\mu_{2}\;$= 1.0; $k_{1}\;$= 0.5, $k_{2}\;$= 0.5, and $k_{3}\;$= 1.0. Our proposed network was implemented with a PyTorch framework and trained with four NVIDIA GeForce GTX1080 Ti GPUs.

### 4.3. Comparison to Other Methods

To evaluate our method, we compare its performance with methods from related work. Table 2 shows the experimental results on the MEVD dataset.

The benchmark network is the prior method of Prétet et al. [2], MLP is used as the common space and optimized with triplet loss. They use ImageNet to extract video features and utilize OpenL3 [40] to extract music features. The dual stream emotional network [6] only pre-trained the network on the emotional dataset, and does not further extract emotional information. In our work, these two classic networks are called content-based baseline networks (B-Content) and emotion-based baseline networks (B-Emotion). Among them, ${\mathrm{DPVM}}_{\mathrm{Sp}}$ is DPVM using splicing fusion strategy, and ${\mathrm{DPVM}}_{\mathrm{In}}$ is DPVM using interactive fusion strategy. Meanwhile, we replace the metric loss with the polarity penalty metric loss (PPML).

The results show the effectiveness of the DPVM network and the best results so far based on interaction fusion. In addition, replacing the classical metric loss with a polarity penalty metric loss has a significant improvement on the retrieval results.

### 4.4. Ablation Study

In order to verify the validity of the proposed content and emotional messages, ablation experiments are conducted for common spaces. Among them, ${\mathrm{DPVM}}_{C}$ refers to the single content path network, and ${\mathrm{DPVM}}_{E}$ refers to a single emotional path network. Meanwhile, we replace the metric loss with PPML. Table 3 shows the experimental results on the MEVD dataset. The results indicate that the content information path is more efficient than the emotion information path. This is because feature extraction networks are pre-trained on large-scale dataset. However, there currently exists a lack of such pre-trained networks for emotional information. The DPVM network using interactive fusion strategy works best. Moreover, the suggested metric loss of polarity penalty also shows good performance. The experimental results demonstrate the effectiveness of using the encoder–decoder structure as a common representation space for content and the ability of the emotion feature extraction scheme to effectively extract emotion information.

## 5. Discussion and Conclusions

In video–music retrieval work, we emphasize the importance of emotional information by designing a dual path network to integrate content information with emotional information. Due to the difference between content information and emotional information, we have various designs for feature extraction and common representation space. Specifically, the content information is extracted through the pre-trained network, and the content common space of the coding and decoding structure is designed to remove redundant information. The path of emotional information adopts the emotional scheme, and constructs the common space of emotional with the MLP as a structure. We also explored ways to integrate content with emotion. Furthermore, we optimized the classical metric loss function according to the characteristics of the task. Finally, we produce and publish an emotion-labeled video music retrieval dataset: MVED, which we hope will contribute to research in this area. As part of future work, we will continue to conduct in-depth research on the interaction between emotion and content. At the same time, in addition to movie videos, we are also exploring further for user-generated videos.

## Figures and Tables

**Figure 1 sensors-23-00805-f001:**
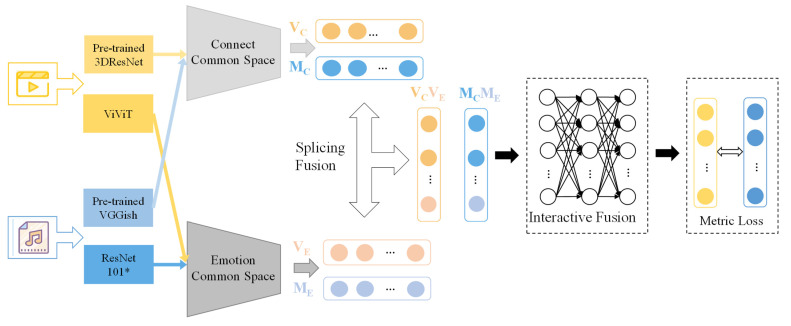
Overview of the proposed DPVM that consists of a content common network, an emotion common network, and a fusion network. Content features obtain their shared representations through the content common space composed of encoder–decoder. Emotional features obtain their shared representations through the emotion common space composed of MLP. Splicing Fusion refers to splicing content features and emotional features of the same modal. Interactive Fusion fully interacts with content information and emotional information. $V_{C}$ is the content feature vector of the video. $\;V_{E}$ is the emotional feature vector of the video. $\;M_{C}$ is the content feature vector of the music. $\;M_{E}$ is the emotional feature vector of the music. * representing ResNet101 with the addition of a channel attention mechanism.

**Figure 2 sensors-23-00805-f002:**
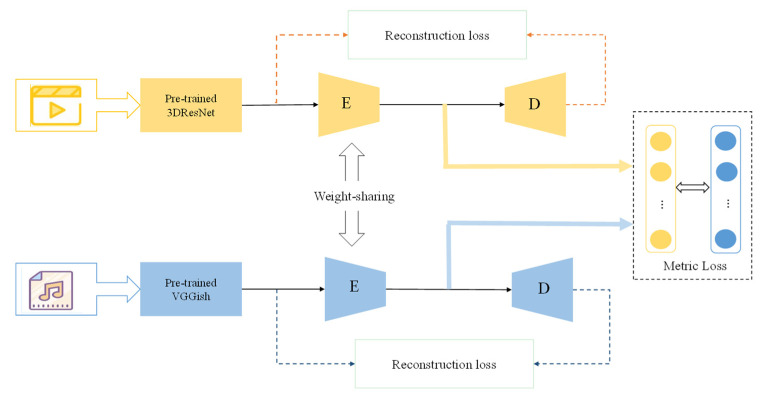
Overview of the proposed content common network. E denotes encoder structure, and D denotes decoder structure. The content common network consists of an encoder–decoder structure as a common representation space in which the weights of the network parameters of the encoder are shared. The content loss include reconstruction loss in the encoder–decoder structure and metric loss between the two modalities of information.

**Figure 3 sensors-23-00805-f003:**
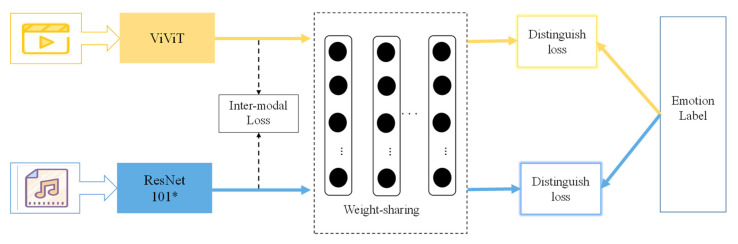
Overview of the proposed emotion common network. MLP as emotional common representation space. The video uses an emotional key frames scheme in the pre-processing stage, * representing ResNet101 with the addition of a channel attention mechanism.

**Table 1 sensors-23-00805-t001:** Example data from MVED.

*Data*	Video Clip Source	Interception Period	Emotional Descriptions	Polarity Labels
100,601	Titanic	01:21:40–01:22:00	Happy	Positive
206,202	The Shining	00:10:37–00:10:52	Depress	Neutral
119,907	Oceans	01:01:50–01:02:05	Nervous	Negative
100,304	Forrest Gump	01:44:42–01:44:57	Happy	Positive
220,405	Stand by Me	01:11:20–01:11:30	Sad	Negative

**Table 2 sensors-23-00805-t002:** Video to music retrieval results on MVED.

*Method*	Recall@1↑	Recall@5↑	Recall@10↑	Recall@15↑	Recall@20↑	Recall@25↑
B-Emotion [6]	5.39	10.61	15.37	20.08	23.53	26.17
B-Emotion [6] +PPML	5.64	11.25	16.80	21.58	24.93	27.33
B-Connect [2]	7.59	15.23	20.31	26.25	30.13	34.27
B-Connect [2] +PPML	7.91	16.43	22.92	27.53	31.29	35.90
${\mathrm{DPVM}}_{\mathrm{Sp}}$	9.13	16.94	22.33	29.50	37.41	42.35
${\mathrm{DPVM}}_{\mathrm{Sp}}$ +PPML	10.42	18.34	24.07	31.74	38.26	43.19
${\mathrm{DPVM}}_{\mathrm{In}}$	10.94	18.84	24.39	34.32	43.26	49.97
${\mathrm{DPVM}}_{\mathrm{In}}$ +PPML	11.53	20.11	26.67	36.01	44.87	50.63

**Table 3 sensors-23-00805-t003:** Video to music retrieval results on MVED.

*Method*	Recall@1↑	Recall@5↑	Recall@10↑	Recall@15↑	Recall@20↑	Recall@25↑
B-Emotion [6] +PPML	5.64	11.25	16.80	21.58	24.93	27.33
B-Connect [2] +PPML	7.91	16.43	22.92	27.53	31.29	35.90
${\mathrm{DPVM}}_{E}$	6.43	13.54	20.26	25.09	31.41	37.26
${\mathrm{DPVM}}_{E}$ +PPML	7.17	14.85	21.44	26.73	32.05	38.16
${\mathrm{DPVM}}_{C}$	8.31	16.42	21.21	27.99	33.94	39.01
${\mathrm{DPVM}}_{C}$ +PPML	9.29	17.86	23.21	28.65	34.30	39.47

## Data Availability

Datasets generated and/or analyzed during the current study are available from https://github.com/GuXin34/DPVM accessed on 31 January 2023.

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
