# Peer review of "A Dual-Path Cross-Modal Network for Video-Music Retrieval"

_sensors, 2023, doi:10.3390/s23020805_

Round 1
Reviewer 1 Report
§ Details of the layers structure of the Pretrained networks are missing. The authors should add the layer details of 3DResNet and VGGish.
§ Line 147: Section title should be after Figure 1.
§ Lines 220 to 228: Some typos and indentation mistakes should be fixed.
§ Line 242: The authors should add the definition of F(Vi) and F(Mi). Also, add the reference for this equation.
§ Line 253: In Figure 3, why you draw two blocks for the “Distinguish Loss”. Why you mentioned the cross-entropy loss by another two different names “Discrimination loss”, “Distinguish Loss”. The authors must make the terms consistent across all the entire manuscript.
§ Line 249 should be moved to under the equation 4 – line 243
§ Line 250: Some typos should be fixed.
§ The authors should add the layer details of ViViT and ResNet 101. You can add their details to a table.
§ Lines 257-262: The advantage of designing the fusion space is not clear.
§ Line 276: the sample mj should be mi
§ Line 295: The authors should evaluate their method performance on the existing recent benchmark: Hong-Im Music-Video 200K (HIMV-200K).

Reviewer 2 Report
1. The abstract section is missing a clear statement regarding the problem solved in this manuscript. The authors still need to highlight the novel ideas proposed in this manuscript.
2. The entire abstract section must be further rephrased as it’s not in a good shape. There a lot of information missing on how you managed to achieve the proposed goals. E.g., “More importantly, we propose a way to combine content information with emotional information.” How was this achieved? What are the novel concepts/strategies proposed in the manuscript.
3. “Hence, automatic retrieval of appropriate music clips based on video information is an urgent issue to resolve.” I would recommend to rephrase or find a more powerful motivation.
4. “We design a metric loss function that is more task-consistent.” The statement is to generic. Based on which idea is this loss function constructed.
5. The manuscript is missing an introduction for section 2.
6. Lines 135 and 137, these is a strange special character between two citations.
7. Section 2 is missing a conclusion on how is the proposed method related to these previous works. What are the different concepts that are proposed in this manuscript compare with these previously proposed concepts.
8. Typos: “(DPVM). The”.
9. Please improve the quality of Figure 1. Please define V_C, M_C, V_E, M_E.
10. Figure 2 is too small. Why are the weights share only for the encoder? What type of network structure is used for the encoder-decoder?
11. Figure 3 is too small. Please add more details about the proposed method.
12. Section 3 is a bit too short. Some details regarding the method are missing. Moreover, all sections are quite small as if they were written for a more compact format.
13. In equation (9), how exactly is the similarity between the data computed? Section 3.5 is not clear. There are also some typos in the equations.
14. Table II must be better formatted. Please avoid multi-page tables. Every second line is empty, therefore it give the feel that the table is half empty as the number are at a too big difference one of each other.
15. Why the comparison with [2] and [6]? Are there no other relevant and more recent articles published?
General comments:
I. The manuscript is a bit too short. The quality of the manuscript must be improved. There are not that many details about the proposed method, and some things are describe maybe too simple.
II. There is also not that much novelty, I feel like these are just some basic ideas that were tested. I would be nice to see some more novel ideas clearly described.
III. Some parts of the manuscript are quite basic.
IV. I think the authors need to work a bit more on the personation and the proposed method.
Reviewer 3 Report
Introduction and related work:
It seems some problems in written English. In literature review, authors are writing like "Yi J et al. [7] proposee a CMVAE model"... and so on. Usually all of these works are done. So better to use proper grammar to express. For example "Yi J et al. [7] proposed the CMVAE...". This should be taken care for the Literature review part.
In introduction also, at the end authors are trying to emphasize on their contributions. What they have done is very encouraging. But when they are writing point no. 1 and 3 seems it is their plan, however it is clear that point 2 they have designed. So this confusion seems to arise in reader's mind about how much they are planning to implement and how much they have done. All this is happened due to writing only. Proper use of tense will remove these confusions.
In reference to Figure 1, by figure it seems video and audio data streams in input. Though mentioning it in diagram itself is advised to increase the readability of Figure.
In reference of Evaluation metric, it is not clear if there is any specific cause due to which only Recall is being used to analyze the system performance.
In discussion and conclusion part, it is visible on line number 404 to 406, "We also further explored..." and then "we optimize"... dissimilarity of tense is creating confusion which has been done and which one will be done...
Round 2
Reviewer 2 Report
- The introduction in section 2 needs to introduce the two subsections and provide a motivation in why they were introduced.
- Figures 1, 2, and 3 are a bit too big.
- Figure 2 caption, please add a few details (1-2 sentences) about the figure so that read does not need to search in the text for the figure description.
- Point 14 in the first review was not addressed. “14. Table II must be better formatted. Please avoid multi-page tables. Every second line is empty; therefore, it gives the feel that the table is half empty as the number are at a too big difference one of each other.”
- Same comment as above for Table III.
- Please check again the manuscript, and provide a better formatting. Some figures are a bit too big, there are extra empty lines (161, 162, 172-175), formatting problems (line 67), the equation numbers are not aligned to the right (lines 295, 298, 306, 308, 315, 318, 320), the tables are poorly designed and contain many extra lines.
